# Could Belief in Fake News Predict Vaccination Behavior in the Elderly?

**DOI:** 10.3390/ijerph192214901

**Published:** 2022-11-12

**Authors:** Vilmantė Pakalniškienė, Antanas Kairys, Vytautas Jurkuvėnas, Vita Mikuličiūtė, Viktorija Ivleva

**Affiliations:** Institute of Psychology, Faculty or Philosophy, Vilnius University, 01513 Vilnius, Lithuania

**Keywords:** older adults, vaccination behavior, fake news, overall beliefs, discernment

## Abstract

Willingness to get a vaccine was important during the COVID-19 pandemic. Previous studies suggest that vaccine hesitation during the pandemic could have been related to truth discernment, belief in information, exposure to misinformation, attitudes to vaccines, and conspiracy beliefs. Previous studies were mostly with younger adults, and studies with older adults are lacking. This study aimed to analyze the relationship between the trust or belief in fake online news (print news was not included), truth discernment, attitudes, and willingness to be vaccinated during the COVID-19 pandemic while controlling for some significant factors/variables that could affect vaccination in a sample of older adults. There were 504 pre-retirees and retirees participating in this study. Participants from Lithuania age ranged from 50 to 90 years old (M = 64.37, SD = 9.10), 58.3 percent were females. Results from several path models predicting the participants willingness to get a vaccine suggested that stronger conspiracy beliefs and skeptical attitudes toward vaccination would be related to lower willingness to get vaccinated. Participants who disbelieved in the headlines were already vaccinated. Therefore, it seems that discernment (the ability to distinguish which information is true and which is not) is not related to the willingness to vaccinate.

## 1. Introduction

The COVID-19 pandemic brought into the picture the importance of vaccines. Even though it has been shown that various vaccines are effective and could reduce symptoms or mortality rates of certain diseases, vaccination rates have not increased drastically over the last decades. In contrast, there is evidence that vaccination rates are decreasing [1]. Many parents are not willing to vaccinate their children in preference for natural immunity [2]. Thus, society, regarding its relation to vaccination, is not homogenous. Many factors could affect a person’s intention to get vaccinated. Previous studies have suggested that the sociodemographic factors are rather significant. According to various studies, men are more likely to be vaccinated than women [3,4]. There is also evidence that ethnicity is related to vaccination [5]. Age is considered as an essential factor in vaccine acceptance. Studies show that older people like to be vaccinated more than younger ones [3,6]. However, vaccination rates during COVID-19 among the 65+ age population have been found to fall below the World Health Organization’s target of 75% in the U.K. [7], and in Singapore, the elderly were less likely to be vaccinated with the COVID-19 vaccine than young people [8]. Studies with older adults showed that their willingness to be vaccinated could also be related to the vaccine characteristics [9,10], their perceived risks [11,12,13], their health-related behavior [11,14], and other factors. However, studies on the elderly suggest that two factors could be especially significant in the decision to be vaccinated or not: attitudes and information.

Both theoretical predispositions and empirical data link attitudes and behavior. The well-known theory of planned behavior [15] postulates that attitude together with the subjective norm and perceived behavioral control leads to intentions, which lead to behavior. Many studies conducted in the context of the theory of planned behavior or outside its reach have confirmed the link between attitudes toward vaccination and vaccination behavior. For example, examining human papillomavirus vaccine uptake in young adult women found that attitudes, as predicted by the theory of planned behavior, were related to vaccination behavior through intentions [16]. Studies conducted in the context of COVID-19 vaccination [17,18], childhood vaccination [19], H1N1 (swine flu) vaccine [20], MMR vaccine [21], etc. have confirmed the link between attitudes and vaccination intentions/behavior. Studies also show that both attitudes and conspiracy beliefs (which could be seen as a particular set of attitudes) are related to vaccination behavior. It is already known that coronavirus and vaccine conspiracy beliefs are related to higher hesitancy in getting a vaccine [22]. Thus, conspiracy beliefs or mentality might be tightly associated with information that a person is exposed to.

Another factor that is important for vaccination is the information. During the COVID-19 pandemic, the search for information was crucial and enormously increased [23,24]. Additionally, during the pandemic, we all faced an “infodemic”. “Infodemic” is often used to describe the flow of a vast amount of information that includes false and fake news during the outbreak of a particular disease [25]. During the pandemic, there was much misinformation, and various conspiracy theories were presented not only about COVID-19 but also about vaccines. According to the World Health Organization [25], an “infodemic” could confuse society, and mistrust in authorities and health care specialists affect people’s health-related behaviors. For example, people could even harm their health because of the decision not to get a vaccine. Much of the information presented in the media, especially on social networks (unless official profiles of health care providers) during the pandemic, was fake or lacked accurate information [26,27]. People like to share and spread novel information, which in many cases is false news, more than conventional information [28]. From the previous literature, it is already known that trust in the information presented in the media could be an essential factor in choosing to be vaccinated. It has been shown in the literature that exposure to negative information and conspiracy beliefs in the media and social media is related to the lower acceptance of vaccines [29,30]. However, information from health care providers, disease control centers, and scientists has been related to a higher willingness to be vaccinated [31]. Thus, it seems that aside from many factors, exposure to information in the media is an essential factor. It is already known that exposure to fake and conspiracy belief information lowers the intention to get the COVID-19 vaccine and the wish to get vaccinated to protect others [32]. It was also found that misinformation presented as evidence-based was related to a lower willingness to get vaccinated and also that females were less robust to misinformation when considering vaccination [32]. A recent study suggests that exposure to fake news, even for a short time (less than 5 min), can modify or change behavior [33]. A researcher has suggested that this change will not be conscious [30]. Consequently, conspiracy beliefs, trust in information, and news online that people are exposed to for a short time could affect their vaccination behavior.

Studies [34,35,36] have hypothesized that attitudes are the antecedents of belief in fake news. For example, it was found that attitudes toward vaccination affected the perceived credibility of fake news and self-expressed motivation to share the information [35]. However, a study on the media portrayals of immigrants found that the opposite direction of the link is also plausible: it was found that media portrayals of immigrants affected attitudes toward immigrants [37]. This led to the hypothesis that exposure to fake news (significantly prolonged exposure) in one way or another could be related to a person’s attitudes and, subsequently, behavior. Many people during the pandemic were exposed to the same information, however, people were not homogenous according to their vaccination behavior. It might be that not only the exposure or belief in the presented information could be an essential factor.

According to Pennycook and Rand [38], there are two ways to evaluate belief in authentic or fake news: one to look at overall beliefs and another to assess the person’s discernment or accuracy to detect actual and false information and content. It is known that general beliefs do not affect people’s ability to tell what is the truth and what is not. Previous studies on COVID-19 fake news have evaluated only the overall belief in information. However, discernment could also be an essential aspect because it is known that discernment is related to poor reasoning, lack of argument, and more deliberation [38], which could affect the decision to be vaccinated. Previous research suggests that discernment for misinformation is influenced by previous knowledge, for example, political knowledge and the evaluation of political information or scientific knowledge while evaluating health-related information [36]. People could fail to discern the truth from false or fake news because they do not reflect their knowledge about a specific topic, especially when they do not have time to stop and reflect and rely on intuitive reasoning. One study suggested that detecting what is fake and what is not fake could be related to vaccination behavior [39]. From the viewpoint of dual-process theory [38,40], it could be concluded that system 1 (autonomous and intuitive) processes are more involved in believing fake information than system 2 processes (deliberative and analytic). The previous research, conducted using dual-process theory, has shown that shorter response time or additional mental load may lead to system 1-based processing [41,42]. Bago and colleagues [43] demonstrated that deliberation (which means more involvement of system 2) leads to less belief in fake political news. Thus, it could be that the shorter time given for processing information could lead to worse discernment, leading to greater acceptance of conspiracy theories or anti-vaccination attitudes and willingness to get vaccinated.

Taken together, hesitation to get vaccinated during the COVID-19 pandemic could have been related to truth discernment, belief in information, exposure to misinformation, attitudes to vaccines, and conspiracy beliefs. Previous studies have mostly been conducted with younger adults, and studies with older adults are lacking. These studies have shown that older adults tend to believe fake information more often than younger people [44]. Even though a recent study suggested that older people are not as susceptible to fake news until a very old age [45], studies evaluating how belief in fake news could affect elderly vaccination behavior are lacking [8]. This study aimed to analyze the relationship between the trust or belief in fake news, truth discernment, attitudes, and willingness to be vaccinated during the COVID-19 pandemic while controlling for some significant factors/variables that could affect vaccination in a sample of older adults.

## 2. Materials and Methods

### 2.1. Participants

There were 504 pre-retirees and retirees participating in this study. The participants’ age ranged from 50 to 90 years old (M = 64.37, SD = 9.10). Participants were selected using a quota sampling procedure. Only participants who were using the Internet were invited to participate in the study. Participants represented the population of Internet users (not the whole population) in Lithuania by age, gender, and living location. However, the proportions of participants according to their age group and gender represent the Lithuanian population. Two hundred and ten invited participants (41.7%) were males, and 294 (58.3%) were females. A total of 474 (94%) participants had Lithuanian nationality (2.6% Russian, 2.4% Polish, and 0.2% Belarussian nationality), and 236 (46.9%) lived in big cities, 174 (34.5%) lived in main district towns or small towns, and 94 (18.7%) in villages. A total of 309 (61.3%) participants had a university or similar education, 329 (65.3%) participants were married or living with a partner, 18 (15.9%) were widows, 147 (29.2%) lived alone, and 251 (49.8) lived with a spouse. Two hundred and twenty-two (44%) participants were officially retired; however, nine were still working full-time, and two worked part-time. A total of 196 (38.9%) participants were working full-time, and 137 (27.2%) participants did not have any children.

### 2.2. Procedure

The data were collected online in the spring of 2021 (during the second lockdown due to the COVID-19 pandemic). Participants were recruited by a company of sociological research from a research participants’ pool. Furthermore, the recruited participants were directly invited to participate in the study. There were 530 participants invited to participate in the study. From them, 504 agreed to participate and participated online when the survey was open. All of the participants’ answers to the presented questions were considered valid.

We developed a list of 10 news headlines; six presented fake information, and four gave accurate information. Five showed health-related or COVID-19-related information, and five presented political information unrelated to COVID-19. We selected headlines that were COVID-19-related and those not related to COVID-19. Some non-COVID-19-related headlines were related to health information because our goal was to look at trust in health news in general, not specifically in COVID. We also decided to include political headlines due to the situation in all countries where the vaccination process was strongly politized [46], as in many EU countries. All of the headlines were fact-checked and all came from national media or social networks. In the beginning, the researchers selected 33 possible headlines/messages, the truth of which could be confirmed or denied based on fact-checking pages in Lithuania (for example, in the fact-checking section of 15 min.lt) or abroad (for example, snopes.com). Researchers read the messages and voted on the reliability of the messages, whether their truth or falsity was clear, and whether the message was readable and engaging. The researcher selected fourteen headlines for the pilot study. One hundred and seventy people (convenience sample) participated in the pilot study, of which 118 (69.8%) were women, and 51 (30.2%) were men. Age ranged from 19 to 68 years (M = 33.8; SD = 9.54). The survey was conducted online. After the pilot study, only ten headlines were left due to understanding the presented headlines by the participants.

Headlines were presented in the format of one of the biggest national media sites or Facebook posts. Each headline had a picture corresponding to the headline and one sentence accompanying the headline. After each headline, participants answered a few questions about the headline, for example, the probability that the information in this headline is accurate. The Latin-square design was used to divide participants into groups according to headlines showing the time and presentation of headlines (Facebook post or media site). Headlines were programmed to be shown for 7, 10, or 15 s or without any time limit. Considering that there was a lockdown in the country, each participant had to evaluate all of the headlines and answer all of the questions online without any researcher present. After the data collection, it was presented to the participants what information was true and what was fake. The local ethical committee approved the study before the data collection.

### 2.3. Measures

Overall belief in headlines. All of the participants read true and false news headlines taken from social media. To evaluate how much the participants believed that the presented headline was accurate, after each headline, all of the participants were asked one question: “What is the probability that the information in this headline is true?” by rating the probability on a 6-point scale ranging from (1) there is a very high probability that it corresponds to reality to (6) there is a very high probability that information is not true. Considering that ten headlines were presented to each participant, the final score of the overall belief in headlines was obtained from the values of 10 items by averaging them. The mean values ranged from 1 to 5.88 (M = 3.75, SD = 0.79). A higher score represents a higher disbelief in the headlines. Since five headlines presented health-related or COVID-19-related information and five presented political information, the belief in the political headlines and health-related belief were calculated separately. The mean values of the political headline beliefs ranged from 1 to 6 (M = 3.37, SD = 0.91) and the mean values of the health-related headline beliefs ranged from 1 to 5.80 (M = 4, SD = 0.68). A higher score represents a higher disbelief in political or health-related headlines. Belief in political and health-related headlines was correlated *r* = 0.31, *p* < 0.001. Discernment. Discernment measures the ability to discern fake from true news (difference in accuracy judgments between true and false news/headlines). On the other hand, overall belief is a measure of bias toward believing any news despite it being true or false. To evaluate how much the participants could distinguish between true and false headlines, we used the same question as for the overall belief in headlines: “What is the probability that the information in this headline is true?” by rating the probability on a 6-point scale ranging from (1) there is a very high probability that it corresponds to reality to (6) there is a very high probability that the information is not true. Six headlines presented fake information, and four presented true information. This was calculated as believing in fake news minus believing in true news (as in [38]). The discernment values ranged from −2 to 27 (M = 13.28, SD = 4.83). A higher score represents higher discernment (better ability to distinguish between true and fake headlines). Again, we calculated the discernment separately for the political headlines and health-related headlines. The political headlines’ discernment values ranged from −3 to 16 (M = 5.24, SD = 3.03). The discernment values of the health-related headlines ranged from −3 to 16 (M = 8.03, SD = 3.28). A higher score represents better distinguishing between true and fake political or health-related headlines. Abilities to distinguish between true and fake political and health-related headlines were correlated *r* = 0.17, *p* < 0.001.

Willingness to get a vaccine. All of the participants were asked if they would get a vaccine against coronavirus (COVID-19) infection. There were four answer options: (1) I will not get a vaccine; (2) I will get a vaccine only if the vaccine is free; (3) I will get a vaccine even if there is a need to pay for that; and (4) I am already vaccinated.

Vaccination attitudes. Participants were asked to complete a 12-item Vaccination Attitudes Examination (VAX) scale [47]. Each item of this scale is rated on a 6-point Likert-type scale ranging from (1) strongly disagree to (6) strongly agree. The VAX scale included four subscales: mistrust of vaccine benefits, worries about unforeseen future effects, concerns about commercial profiteering, and preference for natural immunity. Each subscale included only three items. For this study, we used a composite score of all of the items and did not use individual subscales. The Cronbach’s alpha of the scale was 0.92 (M = 3.27, SD = 1.18). Higher scores on this scale reflect a more negative attitude toward vaccination.

Conspiracy belief. We used five questions in the conspiracy belief questionnaire [48]. This questionnaire measures generic conspiracy beliefs. Each item of this questionnaire is rated on an 11-point Likert-type scale ranging from (0%) certainly not to (100%) certain. The Cronbach’s alpha of the scale was 0.87 (M = 6.48, SD = 2.09). Higher scores on this questionnaire reflect more conspiracy beliefs.

### 2.4. Data Analysis

IBM SPSS 27 and Mplus 8.2 [49] were used for the data analysis in this study. With Mplus software, we tested several path models predicting the participants’ willingness to get a vaccine. Considering that attitudes toward vaccines could be an important predictor of getting a vaccine or not, we first tested whether conspiracy beliefs and attitudes toward vaccines could predict if a person will be or already is vaccinated or not going to get a vaccine. Consequently, we tested whether belief in the headlines and discernment could predict the participants’ willingness to vaccinate. We also tested separate models for the political headlines and health-related headlines. The last model included conspiracy beliefs, attitudes toward vaccines, beliefs in headlines, and discernment while predicting the willingness to vaccinate. We controlled for gender, age, and headlines, seeing time in all the tested models. Model fit was evaluated based on several fit indices: root mean square error of approximation (RMSEA), comparative fit (CFI), and Tucker–Lewis (TLI) index. Based on common practice, the model is considered well-fitting when the RMSEA is smaller than 0.06 (still acceptable fit if it is smaller than 0.08) and CFI and TLI are close to 0.95 or greater [50]. Results are considered significant at *p* ≤ 0.05.

## 3. Results

During the data collection, the country’s vaccination process has already started. Thus, some senior people were already vaccinated at that time. In this sample, 76 (15.1%) participants said they would not get a vaccine, and 275 (54.6%) were already vaccinated during the data collection. A hundred participants (19.8%) mentioned that they would get a vaccine if it were for free, and 53 (10.5%) would get a vaccine even if there were a need to pay. In this study, we created three groups of participants: (1) people that would not get a vaccine—76 participants; (2) people that were willing or at least thinking of getting a vaccine—153 participants; and (3) people that had already received a vaccine—275 participants. The mean values of the variables used in the model—beliefs, discernment, attitudes towards vaccination, and conspiracy beliefs—were tested between these vaccination groups. The results in Table 1 suggest that people who were not willing to get a vaccine had lower levels of overall skeptical beliefs in the headlines; thus, they believed in various headlines more, especially in the political headlines, than the people who were already vaccinated. These people also had a lower ability to distinguish between true and fake health-related headlines compared to people thinking of getting a vaccine. Additionally, people who are not going to get a vaccine had a very skeptical attitude toward vaccination and had the highest conspiracy beliefs. Additionally, the results showed that vaccinated people and people thinking of getting a vaccine were similar in comparing the mean values.

We also evaluated how beliefs, discernment, vaccination attitudes, and conspiracy beliefs are related. Table 2 shows the correlations between the variables presented. Data presented in the table suggest that overall skeptical beliefs are related to the ability to distinguish between true and false information (*r* = 0.33, *p* < 0.001). It also seems that higher disbelief is related to a less skeptical attitude toward vaccination and lower conspiracy beliefs. Or, to put it another way, higher belief is related to a more skeptical attitude toward vaccination and higher conspiracy beliefs. However, higher discernment is related to only less skeptical beliefs about vaccination while skeptical vaccination attitudes are related to conspiracy beliefs.

Considering that participants were presented with headlines in the format of the website (of the national media sites) or Facebook posts, we tested whether these groups differed in beliefs, discernment, vaccination attitudes, and conspiracy beliefs. The results suggest that there were no significant differences between overall beliefs (*t* = −0.08, *p* = 0.938), political headlines beliefs (*t* = −0.05, *p* = 0.961), health-related headlines beliefs (*t* = 0.43, *p* = 0.668), discernment (*t* = 0.33, *p* = 0.740), political headlines discernment (*t* = 0.28, *p* = 0.781), health-related headlines discernment (*t* = 0.23, *p* = 0.818), attitudes toward vaccination (*t* = 1.01, *p* = 0.316), or conspiracy beliefs (*t* = −1.14, *p* = 0.255). There were random groups based on headline exposure time (7, 10, 15 s, and no time limit). Thus, we also looked at the differences between different headline seeing time groups in their beliefs or discernment. The results presented in Table 3 show that there are certain differences between groups in terms of the general beliefs, health-related headlines beliefs, discernment, and political headlines discernment. This suggests that a shorter seeing time (7 s) could result in stronger belief and lower discernment. If a person has no time to delve into the headline, there is a tendency for that person to believe the information that is presented and would treat it like true information. Thus, seeing time could be important in believing and deciding which information is true. However, seeing the format is not an important factor.

Since the previous literature suggests that attitudes toward vaccines could be an essential predictor of getting a vaccine, we tested whether conspiracy beliefs and attitudes toward vaccines could predict if a person will be or already is vaccinated or not going to get a vaccine. Attitudes toward vaccines and conspiracy beliefs were evaluated together with the presented headlines during the pandemic. We assumed that attitudes toward vaccines and conspiracy beliefs were affected by the context that existed back then in the media and society. Thus, although participants were asked about their attitudes to vaccines unrelated to a specific vaccine, we assumed that many thought mainly about COVID-19 vaccines. In this model, as in all other models, we controlled for gender, age, and headline seeing time (as there were differences in beliefs and discernment). The model fit information is presented in Table 4. According to the logistic regression model, it seems that if a person has stronger conspiracy beliefs (OR = 1.28, *p* = 0.003) and a higher skeptical attitude toward vaccination (OR = 4.70, *p* < 0.001), there is a tendency that the person would not be willing to get a vaccine compared to an already vaccinated person. The results also suggest that if a person has a higher skeptical attitude toward vaccination (OR = 1.44, *p* = 0.019), there is a tendency that this person would be in a group that is willing to get a vaccine but not vaccinated yet compared to an already vaccinated person. The model also suggests that persons of older age are related to being in a group where people are already vaccinated. However, gender and headline seeing time were not related to vaccination groups.

In a separate model, we also tested if overall beliefs in headlines and discernment could predict participants’ willingness to vaccinate. Seems that higher overall disbelief in headlines lowers the chances for people to be in non-vaccinated (OR = 0.38, *p* < 0.001) or still thinking about vaccination groups (OR = 0.67, *p* = 0.002). Thus, results suggest that participants who disbelieve in headlines are already vaccinated. Seems that discernment is not related to willingness to vaccinate. Again, this model suggests that persons’ older age is related to being in a group where people are already vaccinated. Gender and headline seeing time were not related to vaccination groups. Taking into the model only political context-related headlines–beliefs into them and discernment–it seems that disbelief in political headlines lowers the chances for people to be in the non-vaccinated (OR = 0.57, *p* < 0.001) group only. Thus, people who believe in political headlines are not willing to get a vaccine. Again, a person’s older age is related to being in a group where people are already vaccinated. Gender and headline seeing time were not related to concrete vaccination groups. Taking into the model only the health-related headlines—beliefs into them and discernment—it seems that disbelief in health-related headlines and decrement is unrelated to willingness to vaccinate. Only the older age of persons was related to being in a group where people were already vaccinated. Considering that people were viewing the political and health-related headlines simultaneously, we entered into the model both the political and health-related headlines’ beliefs and discernment. Results from such a model suggest that only higher disbelief in political headlines lowered the chances for people to be in the non-vaccinated (OR = 0.58, *p* < 0.001) group compared to vaccinated people. Putting together the results from all of the models, it seems that disbelief in headlines (primarily political), attitudes toward vaccination, and conspiracy beliefs are essential for willingness to vaccinate. The correlations in Table 2 also indicated no relationships between discernment and vaccination groups. Models including discernment also suggested that discernment cannot predict a person’s willingness to get a vaccine. Thus, discernment could affect willingness to vaccinate via other variables such as vaccination attitudes or conspiracy beliefs.

In the final model, we put all the predicting variables that were significant predictors in previous models together: vaccination attitudes, conspiracy beliefs, headlines beliefs, and discernment. The results from this model were similar to previous models. They suggest that if a person has a higher skeptical attitude toward vaccination (OR = 4.41, *p* = 0.001) and also higher conspiracy beliefs (OR = 1.28, *p* = 0.012), there is a tendency that this person would not be willing to get a vaccine compared to an already vaccinated person. The results in the final model also suggest that higher overall disbelief in the presented information lowers the chances for the participant to be still thinking about the vaccination group compared to the already vaccinated participants (OR = 0.68, *p* = 0.008). Thus, the results imply that participants who disbelieve headlines are already vaccinated.

## 4. Discussion

The COVID-19 pandemic raised the importance of vaccination in every country again. Although many people were dying or had severe conditions during the COVID-19 pandemic, vaccines that could save lives were not readily accepted by people in many countries [51]. Vaccine hesitancy is a common phenomenon in many countries [1]. Over the last few years, many researchers [3,10,11] have been trying to figure out why people were hesitant to vaccinate. Aside from the sociodemographic factors, the previous literature suggested that two factors could be significant in the decision to vaccinate or not in the elderly: attitudes and adequacy of receiving information. However, these factors were primarily evaluated separately from each other. In this study, we assessed the relationships between attitudes, trust, or belief in the presented information, ability to detect what information is accurate and what is not, and the willingness to be vaccinated during the COVID-19 pandemic in a sample of older adults in Lithuania. As in previous studies, we started with univariate analysis. However, multivariate models were constructed to analyze the complex relationships between constructs.

In the univariate analysis, three participant groups were compared: vaccinated, getting a vaccine, and not getting a vaccine. People in the non-vaccinated group had the highest skepticism toward vaccination, conspiracy beliefs, the lowest level of distrust in the information, especially in political information, and the lowest discernment level in health-related information. Thus, people who were not willing to get a vaccine trusted the presented information more; however, they were more skeptical and had conspiracy thinking. For them, it was hard to distinguish which health-related information was accurate. In contrast, those already vaccinated had the highest disbelief in information, especially political ones. This result is in contrast to studies that found relationships between high trust in information from government bodies, trust in political bodies [52], or trust in the health care system [53] and vaccination. In this study, we did not ask participants about trust in governmental bodies or the health care system. The information presented was in the form of a Facebook post or national media site reports, thus not as official governmental information. Considering that vaccinated people have a less skeptical attitude toward vaccines, it might be that those people could have their own stronger opinion and do not trust only the presented headlines and information they could read. An alternative is that people figured out that we presented some fake and some accurate information and understanding that mostly answered that they did not believe it. However, it is interesting that trust in political information/headlines could be related to a weaker willingness of people to do what the government has asked them to behave. It is also known that people choose to get vaccinated or not according to their political views—Republican voters had a significantly lower vaccination rate [54]. In this study, we did not evaluate the political views of the participants. However, it might be that the presented political information was related or not to their political views and, in that case, more strongly related to vaccination behavior.

We also tried to predict what factors could affect whether people were already vaccinated. Analysis of the literature led us to hypothesize that hesitation to get a vaccine during the COVID-19 pandemic could be related to truth discernment, belief in information, vaccine attitudes, and conspiracy beliefs. Previous authors [32,39] have tested these factors separately. However, all of the more complex models tested in this study suggest that taking all the factors together—beliefs in the news, attitudes toward vaccination, and conspiracy beliefs—are essential for vaccination behavior.

In this study, we conceptualized belief in the news in two ways, discernment and overall beliefs, as suggested by Pennycook and Rand [38]. In our analysis, considering both discernment and overall beliefs, we found that only overall beliefs would predict vaccination behavior. The results suggest that participants who disbelieved in the news are more likely to already be vaccinated. Previous research has shown that exposure to negative information and conspiracy beliefs in the media and social media was related to a lower acceptance of vaccines [29,30]; therefore, overall skepticism or disbelief could be a protective factor and lead to a better acceptance of vaccines. Moreover, the analysis of health and politics-related news showed that overall beliefs in health-related information and discernment of this information were unrelated to vaccination. Only beliefs in political information were related to vaccination. It seems that trust/belief in political information is crucial for such types of behavior. During the pandemic, many political or governmental bodies provided health-related information to society. It could be that people associate health-related information with political information because politicians provide such types of information. Additionally, trust in vaccination and political information was related during vaccination. Furthermore, people did not always trust vaccines in the same way as they did not trust political information. In these cases, trust in political information became a more critical factor in this sample.

It was hypothesized that better discernment, especially in health-related information, would increase the chances of being in the vaccinated group. It is known that better discernment is related to better cognitive functioning [38] and relates to previous knowledge of a specific topic [38] or scientific knowledge in the COVID-19 case [36]. Thus, it might be that people participating in the study had very similar cognitive abilities and did not have that much variation in between. Participants in the study might have had similar exposure to information and similar knowledge about the presented health-related or political information. It also could be that discernment cannot directly predict vaccination behavior. It was found that discernment affects the trust of belief in the news [55], and trust could be related to vaccination behavior, as our data support. Thus, the mediation or moderation effects should be considered aside from testing the direct effects.

Considering that attitudes toward vaccination are essential for vaccination behavior, one predicting factor was attitudes. A more skeptical attitude toward vaccinations leads to being in non-vaccinated groups: going to get a vaccine and not going to get a vaccine. Previous studies, especially with younger participants, suggested that attitudes toward vaccination and conspiracy beliefs (a unique set of attitudes) are related to vaccination behavior [18,29]. We found that the same relationships are also in the older adult sample. In previous studies published lately, attitudes toward vaccination or conspiracy beliefs were mainly related to COVID-19. We found that asking questions unrelated to COVID-19 relationships would be similar. It seems that it is crucial to build positive attitudes toward vaccinations on the whole for society’s immunization to decrease vaccine-preventable illnesses. We should also agree that during the pandemic, thoughts about COVID-19 affected the attitudes and beliefs. Thus, even by asking more global questions about vaccination, people might think about the existing situation.

In this study, we controlled for the demographic factors of age and gender. Age is considered as an essential factor in vaccine acceptance. In this study, we found that persons’ older age was related to better vaccination, which means that older people were in a group where people were already vaccinated. Previous studies have suggested that older people were more willing to be vaccinated than younger people [52,56]. However, some studies have suggested that younger people were more willing to be vaccinated against COVID-19 [57]. In previous studies, there were adults of various ages (starting from 18); thus, older adults were compared with very young people. Our study found that older adults 50 years and older had differences, and older age would be related to a higher willingness to vaccinate. Previous studies with older adults showed that willingness to be vaccinated was related to their perceived risks [11] or their health-related behavior [14]. Considering that older age is related to worse health, older people are expected to be more willing to get a vaccine. We did not control for health status in this study. Thus, we could only assume that the 50 year old people in our sample might have better health and were not in the vaccinated participants’ group. Even though the male gender was related to better vaccine acceptance than females [57] in previous studies, we did not find that gender could be related to better or worse vaccination. It might be that the COVID-19 vaccination was important for all people to survive and not get sick, unlike other vaccines presented in previous studies.

We based our experiment on a dual-process theory [40]; therefore, we controlled the exposure time of the news. Previous studies on this theory have suggested that a shorter response time or additional mental load may lead to system 1-based processing (more automatic answer) and a greater belief in fake information [41]. In this study, we found similar results suggesting that time matters for believing, as in previous literature. Overall disbeliefs and health-related disbeliefs were lowest in a group that saw the presented information only for 7 s, while previous studies indicated that a shorter time is essential. However, how short it must be was not presented. People with a short response time had stronger beliefs in the presented headlines due to not having time to analyze and deliberate on the information as it should be in the system 2-based processing [38]. Interestingly, their discernment, especially of political-related information, was the lowest. In this case, in all of the models, the time of seeing the headlines was included as a controlling factor. However, it did not affect the models that we tested. This means that no moderation effect was found for the model. Once again, we tested the direct models—all the variables predicting the vaccination group simultaneously. It could be that the seeing time was directly related only to beliefs and discernment, and moderation only with beliefs and understanding should be tested.

This study had some limitations. It was a cross-section design study that allows for the evaluation of the associations between factors. However, we should be careful about the causal relationships. Even though we tested them in this study, they should be replicated with longitudinal data. Longitudinal data could also test the mediation models and the direction of effects; it is possible that belief and discernment could be affected by attitudes toward vaccination or conspiracy beliefs or even vice versa. The sample in this study might have been fairly homogenous. In this study, people using the Internet from home participated because of the lockdown. For this study, a tablet or stationary computer was required to see the presented headlines. Older people might not have a computer (using the Internet on smartphones) at home and use it only at work if they still are working or in public libraries. Thus, some potential participants could not reach this study during the pandemic. From previous studies, we know that discernment is related to previous knowledge [38], cognitive functioning [38], digital literacy [55], and personality traits [58]. In this study, we did not control for any of these factors, which could affect the discernment, beliefs or attitudes, and even vaccination behavior. Aside from these limitations, the results of this study suggest that older age vaccination behavior during pandemics could be predicted by attitudes toward vaccination, conspiracy beliefs, and general beliefs in the information that could be true or false. In the future, it might be helpful to reveal the similarities or differences between the beliefs in fake news, attitudes toward vaccination, and vaccination behavior between young and older people to evaluate whether the relationship could be stable for everyone. Considering this study was conducted in one country only, it would be helpful to test the same ideas in other countries to see if the processes could be the same. Thus, this study suggests that the ability to distinguish which information is accurate and which is not, is not essential for vaccination. This implies that society must build positive attitudes toward various vaccines. The results of this study suggest that vaccination can be promoted by increasing people’s criticality, encouraging them to think longer before reading some information (because the shorter they read, the more they believe). It is essential to look at vaccination not only as a health issue but also as a political issue and to consider the political climate and prevailing conspiracy theories when carrying out promotion campaigns. It also means that belief in political content-related information could be significant. Considering that there is information that is fake or presenting part of the truth and present various politicians’ opinions that sometimes could just be offering conspiracy theories, it is essential to provide objective information in the national media. Going beyond, our study shows that people tend to believe social media posts, even when the source is unclear. Furthermore, this can be used rather than fighting against it. For example, public institutions and the health care system can shift to the spread of less official, verified, and good information on social media that is read and believed by older people. On the other hand, with the increasing digitalization of medicine, some facts, names, and screenshots of official digital health systems can also be used for the production of fake news, which, in turn, can worsen the vaccination or health behavior in society.

## 5. Conclusions

In conclusion, older people who were not getting vaccines during the COVID-19 pandemic had higher belief in the presented information, especially political ones, and had a lower ability to distinguish between accurate and fake health-related information. Additionally, they had a very skeptical attitude toward vaccination and had the firmest conspiracy beliefs. Vaccination behavior during the pandemic was tightly related to peoples’ skeptical attitudes toward vaccination and conspiracy beliefs. It seems crucial to build positive attitudes toward vaccinations for society’s immunization to decrease vaccine-preventable illnesses. Higher disbelief in the presented information, primarily political, was related to the willingness to take the vaccine. Since vaccination processes were strongly politized in many countries, it seems that political, but not health-related, information was essential for the vaccination. Thus, political figures should take this into account. The ability to distinguish what is accurate information and what is not was not related to vaccination behavior. Thus, this study demonstrates that taking attitudes, beliefs, and discernment together, only attitudes and beliefs are essential in predicting older peoples’ vaccination behavior, at least in Lithuania.

## Figures and Tables

**Table 1 ijerph-19-14901-t001:** Differences between the means and standard deviations of variables used in models between vaccination groups.

Variables	Not Going to Be Vaccinated	Going Be Vaccinated	Vaccinated	F	*p*
*Overall beliefs, Mean (SD)*	3.32 (0.80) _a_	3.70 (0.78) _b_	3.90 (0.74) _c_	17.50	<0.001
Political information beliefs, Mean (SD)	3.07 (1.02) _a_	3.30 (0.91)	3.50 (0.86) _b_	7.73	<0.001
Health-related information beliefs, Mean (SD)	3.89 (0.69)	4.01 (0.67)	4.03 (0.69)	1.13	0.324
*Discernment, Mean (SD)*	12.47 (4.46)	13.74 (4.62)	13.24 (5.03)	1.76	0.173
Political headlines, Mean (SD)	5.32 (3.08)	5.20 (2.79)	5.25 (3.16)	0.04	0.965
Health-related headlines, Mean (SD)	7.16 (3.27) _a_	8.54 (3.13) _b_	7.99 (3.32)	4.59	0.011
*Vaccination attitudes, Mean (SD)*	4.52 (0.57) _a_	3.34 (0.84) _b_	3.46 (0.80) _b_	55.49	<0.001
*Conspiracy beliefs, Mean (SD)*	8.16 (1.95) _a_	6.34 (2.04) _b_	6.10 (1.93) _b_	33.25	<0.001

Note. Significant differences are marked in bold. Means with different letters differed significantly in a particular group at the *p* < 0.05 level.

**Table 2 ijerph-19-14901-t002:** Correlations between variables.

Variables	1	2	3	4	5	6	7	8	9
1. Overall beliefs	−								
*2. Political information beliefs*	0.91 ***	−							
*3. Health-related information beliefs*	0.57 ***	0.31 ***	−						
4. Discernment	0.33 ***	0.27 ***	0.34 ***	−					
*5. Political headlines*	0.24 ***	0.26 ***	0.10 *	0.74 ***	−				
*6. Health-related headlines*	0.26 ***	0.16 ***	0.42 ***	0.79 ***	0.17 ***	−			
7. Vaccination attitudes	−0.43 ***	−0.32 ***	−0.20 ***	−0.13 **	−0.01	−0.18 ***	−		
8. Conspiracy beliefs	−0.36 ***	−0.35 ***	−0.04	0.05	−0.07	0.00	0.51 ***	−	

Note. * *p* < 0.05; ** *p* < 0.01; *** *p* < 0.001. 1—overall beliefs, 2—political information beliefs, 3—health-related information beliefs, 4—discernment, 5—political headlines, 6—heath related headlines, 7—vaccination attitudes, 8—conspiracy beliefs.

**Table 3 ijerph-19-14901-t003:** Differences between the means and standard deviations of variables used in models between experimental groups.

Variables	7 s	10 s	15 s	No limit	F	*p*
*Overall beliefs, Mean (SD)*	3.57 (0.82) _a_	3.81 (0.75) _b_	3.79 (0.76) _b_	3.83 (0.80) _b_	3.06	0.028
Political information beliefs, Mean (SD)	3.20 (0.92)	3.44 (0.85)	3.38 (0.90)	3.47 (0.95)	2.34	0.072
Health-related information beliefs, Mean (SD)	3.86 (0.68) _a_	3.96 (0.68)	4.11 (0.64) _b_	4.08 (0.72) _b_	3.57	0.014
*Discernment, Mean (SD)*	12.05 (4.89) _a_	13.60 (4.73) _b_	13.36 (4.53) _b_	14.10 (4.99) _b_	4.22	0.006
Political headlines, Mean (SD)	4.50 (2.96) _a_	5.63 (2.66) _b_	5.28 (2.99) _b_	5.56 (3.39) _b_	3.75	0.011
Health-related headlines, Mean (SD)	7.55 (3.38)	7.96 (3.69)	8.08 (3.04)	8.54 (2.91)	1.97	0.118
*Vaccination attitudes, Mean (SD)*	3.64 (0.84)	3.66 (0.95)	3.71 (.87)	3.68 (0.78)	0.14	0.939
*Conspiracy beliefs, Mean (SD)*	6.30 (2.04)	6.56 (2.20)	6.58 (2.22)	6.45 (1.91)	0.49	0.688

Note. Significant differences are marked in bold. Means with different letters differed significantly from another group with a different letter(s) at the *p* < 0.05 level. Means with the same letters differed in between.

**Table 4 ijerph-19-14901-t004:** Model fit information.

	N of Parameter	Loglikelihood	AIC	BIC
Vaccination attitudes and conspiracy beliefs	12	−402.52	829.03	879.70
Headlines beliefs and discernment	12	−442.25	908.51	959.19
Political headlines beliefs and discernment	12	−451.47	926.93	977.61
Health-related headlines beliefs and discernment	12	−453.75	931.49	982.16
Political and health-related headlines beliefs and discernment	16	−447.51	927.01	994.57
Final model	16	−399.28	830.55	898.11

## Data Availability

The data presented in this study are available on request from the corresponding author.

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
