# Peer review of "Could Belief in Fake News Predict Vaccination Behavior in the Elderly?"

_ijerph, 2022, doi:10.3390/ijerph192214901_

Round 1

Reviewer 1 Report

The paper is overall well written, but there are some concerns particularly regarding the method sections that must be addressed. Below are some issues that can improve the paper;

1. For selection of the headlines any content validation steps were adopted? If yes please explain (face validity and content validity by sending them to the experts to get feedback)

2. Onward from Line 459 please delineated some theoretical implications of this study and relate how it is different or align with prior literature.

3. Please clarify some managerial implications of this manuscript in light of mounting concerns about healthcare facilities for old age people beyond this topic relate it with some ongoing phenomenon like digital health consultancy, how these results can help to implement more sophisticated plans for betterment of healthcare facilities for old age group.

Good luck 

Reviewer 2 Report

The manuscript addresses an interesting topic. However, it needs some improvements before publication. The main ones are the following:

On page 2 when talking about information, the authors should contextualize the importance of news consumption and information during COVID-19. To do this the authors can use these references:

https://doi.org/10.3145/epi.2020.mar.23

https://doi.org/10.17231/comsoc.40(2021).3283

https://doi.org/10.30935/ojcmt/120126

It is also recommended that authors use this reference on the spreadability of fake news to contextualize its importance:

https://doi.org/10.1126/science.aap9559

The statement on lines 122-124 must be supported by a reference.

The survey sample is small. The authors should explain whether the composition of the sample is representative in relation to the population of Lithuania (e.g. is the percentage of women in the sample similar to the percentage of women of that age group in Lithuania?). Now this is not done.

Authors should explain why non-COVID-19 related headlines were used in the study.

In section 2.3 the discernment measure should be further explained to ensure that it is different from the Overall belief in headlines measure.

At the end of the paper, the authors should answer the following questions: what implications for public policy do the results have? How could vaccination be promoted based on the findings obtained?

Reviewer 3 Report

It is appropriate to publish the study in this form. On the other hand, I would like to offer a number of suggestions that will improve the study in my opinion:

1. It should be emphasized which country the study is specific to. I would also suggest that the name of the country that is the subject of the study should be included in the study.

2. In the summary section of the study, the findings of the study can be explained more comprehensively.

3. How were the participants invited to the study? According to which sample were the participants selected as the sample of the study? It is stated that the majority of the participants are Lithuanian nationals. And what are the nationalities of the other participants? How many people were invited to the study? How many positive responses were received? All of them? If not, how many people received a negative response? Were the answers of all participants considered valid? If not, how many people's answers were disqualified?

4. It is claimed that the study tried to measure the effect of the relationship between trust or belief in fake news, truth discrimination, attitudes and the desire to be vaccinated on the elderly during the COVID-19 pandemic. But when I read the abstract of the study, I did not understand whether this was done only through internet news. So, did you only study on online news? For example, are print newspaper and magazine news out of the scope of your study? These explanations should be included in the abstract.

5. The limitation of the study was emphasized. On the other hand, the study can make suggestions for future studies in the last section. For example, it may be suggested to reveal the similarities and/or differences between the attitudes of young and elderly people on the subject.

The items mentioned above are just suggestions. So it depends on preference. If writers think they've already done that, they don't need to do it anyway. Best wishes, best regards.
